# Growth Performance, Gut Environment and Physiology of the Gastrointestinal Tract in Weaned Piglets Fed a Diet Supplemented with Raw and Fermented Narrow-Leafed Lupine Seeds

**DOI:** 10.3390/ani10112084

**Published:** 2020-11-10

**Authors:** Anita Zaworska-Zakrzewska, Małgorzata Kasprowicz-Potocka, Robert Mikuła, Marcin Taciak, Ewa Pruszyńska-Oszmałek, Andrzej Frankiewicz

**Affiliations:** 1Department of Animal Nutrition, Faculty of Veterinary Medicine and Animal Science, Poznan University of Life Sciences, Wolynska 33, 60-637 Poznan, Poland; anita.zaworska-zakrzewska@up.poznan.pl (A.Z.-Z.); robert.mikula@up.poznan.pl (R.M.); andrzej.frankiewicz@up.poznan.pl (A.F.); 2Department of Animal Nutrition, The Kielanowski Institute of Animal Physiology and Nutrition, Polish Academy of Sciences, Instytucka 3, 05-110 Jabłonna, Poland; m.taciak@ifzz.pl; 3Department of Animal Physiology and Biochemistry, Faculty of Medicine Veterinary and Animal Science, Poznan University of Life Sciences, Wolynska 35, 60-637 Poznan, Poland; ewa.pruszynska@up.poznan.pl

**Keywords:** lupine, fermented feed, gastrointestinal tract, performance, pigs’ nutrition

## Abstract

**Simple Summary:**

Fermented feed in growing pig nutrition may influence microbiota of the gastrointestinal tract, improve utilization of nutrients from the diet, and reduce the level of excreted ammonia and phosphorus released into the environment. In the research, fermentation of narrow-leafed lupine seeds was provided and fermented seeds were added to the pigs’ diet. In the 28-day experiment, 24 male pigs were divided into three groups. The control group was fed a soybean meal diet, whereas in the experimental diets, 50% of soybean meal (SBM) protein was replaced by raw or fermented lupine seeds. The influence of fermentation on performance results, gut environment and physiology, and selected blood metabolic parameters in young pigs, were analyzed. Fermentation did not affect pigs’ performance, metabolic, microbial and most gastrointestinal tract parameters, but influenced crypt depth, concentrations of short chain fatty acids and p-cresole in the proximal colon segment, and significantly lowered the pH of the middle colon digesta and ammonia contents. Fermentation improved the chemical composition of seeds, but due to the lack of a significant improvement in the performance, the results may prove to be economically unviable.

**Abstract:**

The aim of this study was to: (1) provide controlled fermentation of narrow-leafed lupine seeds; (2) monitor seed composition, and (3) determine the influence of fermentation on the performance, gut environment and physiology, and selected blood metabolic parameters, in young pigs. Firstly, the effect of 24 h lupine seed fermentation by bacteria and yeast on seed chemical composition was determined. It increased contents of crude protein, crude fiber and ash, but reduced nitrogen-free extractive levels. The amino acid profile of fermented lupine (FL) was similar to that of raw lupine (RL) seeds, whereas the contents of oligosaccharides and P-phytate decreased significantly, in contrast to alkaloids. In fermented feed, pH dropped from 5.5 to 3.9. In the 28-day experiment, 24 male pigs were divided into three groups. The control group was fed a soybean meal diet (SBM), whereas in the experimental diets, 50% of SBM protein was replaced by RL or FL. Afterwards, eight pigs from each group were euthanized and their digesta and blood samples were collected. The FL use did not affect pigs’ performance, nor their metabolic, microbial and most gastrointestinal tract parameters, but influenced crypt depth. Fermentation affected concentrations of short chain fatty acids and p-cresole in the proximal colon segment. In the small intestine, the levels of acetate and butyrate decreased, and, in the caecum, the propionate level decreased. Fermentation significantly lowered the pH of the middle colon digesta and ammonia contents compared to RL. A part of SBM may be successfully replaced by RL and FL in young pigs’ diets.

## 1. Introduction

In the last few years, the prices of feed components in the world market, especially high-protein feed, remained 2-fold higher than ten years ago. This growth was due to the rising demand, fluctuations in supply and speculation in commodity markets. Moreover, the whole of Europe, including Poland, for many years reduced its own production of protein components for animal feeds. The need to secure domestic sources for feed production, in case of an unexpected slump in global trade, requires finding alternative protein components, comparable to soybean meal (SBM) in terms of quality and economic viability. Seeds of legumes, especially lupine seeds, can play an important role in this respect. However, according to some studies [1,2,3,4,5], the use of lupine seeds in piglets’ diet should be limited. This is related to the presence of anti-nutritional substances in the seeds. One of the processing methods that can improve the nutritional value of seeds is fermentation. Currently, fermented feed in growing pig nutrition is popular, because it enhances pig performance and health [6,7,8]. Furthermore, it may influence microbiota of the gastrointestinal tract, in particular from the *Enterobacteriaceae* family, and due to the improved utilization of nutrients from the feed, it can reduce the level of excreted ammonia and phosphorus released into the environment [9,10]. In addition, different types of fermented feed have been widely investigated to reduce the use of antibiotic growth promoters [8,11] and decrease feed price by using industrial high protein products in modern swine production [12]. Therefore, fermented feed is recommended due to its high nutritional value and digestibility, but the effects on pig growth performance are inconsistent [13]. So far, the fermentation process has been primarily used for the starch components [6,12,14,15]. However, the treatment of lupine seeds by fermentation to ensure their use as a source of protein for animals has been studied before in preliminary research on rats [16,17] and on pigs [18]. The results showed that fermented lupines are characterized by higher nutritional value and better digestibility of protein and amino acids by animals. So, it was hypothesized that fermentation of lupine seeds would provide new products with an enhanced nutritional value in comparison with unprocessed seeds, and that would, additionally, positively impact the growth, microbial community and physiology of the gastrointestinal tract in pigs. Therefore, the aim of this study was to investigate the chemical characteristics of raw and fermented narrow-leafed lupine seed (*Lupinus angustifolius*, cv. Neptun) and to determine the influence of fermentation on the production results, gut environment, physiology of the gastrointestinal tract and selected blood metabolic parameters in young pigs.

## 2. Material and Methods

### 2.1. Fermentation

Seeds of lupine cv. Neptun obtained from the Polish Plant Station were used. The fermentation process of lupine seeds was carried out according to the method described before by Zaworska et al. [18].

### 2.2. Animals and Diets

All the experimental procedures complied with the guidelines of the Local Ethical Committee. The experiments were approved by the Local Ethical Committee in Poznan and were in accordance with the Resolution No. 43/2011 of 15 May 2011. Pigs received all necessary veterinary vaccinations and had an unlimited access to water and feed.

### 2.3. Experiment on Animals

The experiment was conducted on 24 castrated male weaned piglets (P76 × Naima), aged 35 days with an initial body weight of about 9.6 kg assigned to one of three groups, according to body weight analogues. The animals were kept in individual pens and fed ad libitum with complete feed mixtures for 28 days. Diets (Table 1) were formulated according to the GfE [19] recommendations using the results of the authors’ original lupine analysis and ileal digestibility coefficients of crude protein and amino acids [18]. The control diet contained 28.5% SBM, whereas in the experimental diets 50% of the SBM protein was replaced by raw (RL) or fermented (FL) narrow-leafed lupine seeds (20.0% and 18.0% in the experimental diet, respectively).

The average daily body weight gain (ADBG) and feed intake (FI) were recorded, and, from this, the average feed conversion ratio (FCR) was calculated.

At the last day of the experiment, blood samples were collected from the auricular vein. Serum samples were prepared by centrifugation at 1500× *g* for 15 min at 4 °C and they were stored at −40 °C for further analyses. After the experiment, 6 animals from each group were euthanized. Directly after euthanasia (ca. 10 min), the ileal, caecal and colon digesta were collected for pH measurements and chemical and microbial analyses. Small intestine tissues were sampled for morphometric studies. The ileal and caecal digesta were sampled and frozen (at −20 °C) to determine dry matter (DM), pH, ammonia, viscosity and short chain fatty acids (SCFA). The colonic digesta were sampled from its proximal, middle and distal parts (segments I, II and III, respectively), frozen and analyzed for pH, ammonia, SCFA and phenolic compounds content.

### 2.4. Chemical Analyses

For chemical analyses, the samples were ground to pass through a 0.5-mm sieve. The raw and fermented seeds (n = 4) were analyzed in duplicate for dry matter (DM), crude protein (CP), ether extract (EE), crude fiber (CF), ash (CA) and amino acids using methods 934.01, 976.05, 920.39, 978.10 and 942.05, respectively, according to AOAC [20]. Nitrogen-free extractives (NFE) were calculated: NFE = DM − (CP + CA + CF + EE). Lupine alkaloids were extracted from flour by trichloroacetic acid and methylene chloride (Sigma-Aldrich, Munich, Germany). The determination of alkaloids was conducted using gas chromatography (Shimadzu GC17A, Kyoto, Japan) with a capillary column (Phenomenex, Torrance, CA, USA). Raffinose family oligosaccharides were extracted and analyzed by high-resolution gas chromatography, as described previously by Zalewski et al. [21]. Phytate was assayed according to the method prepared by Haug and Lantzsch [22].

The pH of the digesta was measured using a microelectrode and a pH meter (model 301, Hanna Instruments, Vila do Conde, Portugal). Ammonia was extracted and analyzed by the spectrometric method using Nessler reagent (POCh, Gliwice, Poland). For viscosity determination, the samples were centrifuged at 10,000× *g* for 10 min. and the supernatant was withdrawn and determined using a Brookfield Digital DV-II+ cone/plate viscometer (Brookfield Engineering Laboratories Inc. Stoughton, MA, USA) maintained at 30 °C and at a shear rate of 60 1/s. WEV units are mPas·s (mPas × s = cP = 1 × 100 dyne s/cm^2^).

The SCFA analysis was performed according to the procedure described by Barszcz et al. [23] on an HP 5890 Series II gas chromatograph (Hewlett Packard, Waldbronn, Germany) with a flame-ionization detector and a SupelcoNukol (Supelco, Bellafonte, KY, USA) fused silica capillary column (30 m × 0.25 mm i.d.; 0.25 mm). Helium was used as the carrier gas. The concentrations of individual SCFA were estimated in relation to an internal standard (isocaproic acid) using a mixture of SCFA standard solutions. Phenol, p-cresol and indole concentrations were analyzed based on the method described by Taciak et al. [24] using the Shimadzu GC-2010 gas chromatograph (Shimadzu, Kyoto, Japan) with a flame-ionization detector, a Supelco Nukol fused silica capillary column (60 m × 0.32 mm i.d.; 0.25 µm) and helium as the carrier gas. Phenol, p-cresol and indole concentrations were calculated using the standard curves and in proportion to 5-methylindole as the internal standard.

For morphometric analyses of ileum tissues, parts of the intestines were fixed in 4% formalin, subsequently washed and dehydrated in ethyl alcohol of increasing concentration, X-ray xylene, and then embedded in paraffin. Sections with a thickness of 6–7 microns were cut on a HM 340E microtome (Microm, Munich, Germany) and stained by the HE method. Preparations were analyzed using a Axioimager light microscope. Sections of the correct structure were scanned using a Mirax Desk scanner (Carl Zeiss, Roßdorf, Germany). For measurements of structures, the AxioVision software (Carl Zeiss, Roßdorf, Germany) was used. The scanned sections were stained by the HE method and mucous membrane thickness, the length of villi and crypt depth were measured in six replications.

Samples for microbial analyses were prepared by adding 27 mL of buffered peptone water (Oxoid, Wade Rd, Hampshire, UK) to 3 g of samples and homogenizing for 30 s in a laboratory stomacher. Microbial counts were determined using a decimal dilution series of homogenized samples. The total bacteria count and lactic acid bacteria count were determined by plate standard methods using plate count agar and MRS broth (Oxoid), respectively, after 72 h incubation at 30 °C. The *Salmonella* count was determined using pre-supplemented Dichloran Rose Bengal Chlorcamphenicol and agar Salmonella Chromogen (Oxoid) after 18 and 24 h incubation, respectively, at 37 °C. The yeast content was calculated using pre-supplemented DRBC (Oxoid) after incubation at 25 °C for 3–5 days. The coliform bacteria were determined using Violet Red Bile Lactose agar (Oxoid) after 24 h of incubation at 30 °C.

The metabolic profile was marked using colorimetric assays according to the manufacturer’s instructions. Optical density of samples was determined using a microplate reader (Synergy 2, Winooski, BioTek, Vermont, MA, USA). The level of non-esterified fatty acids (NEFA) was measured using the kit from WAKO (Cat No.: 434-91795 and 434-91995; Wako Chemicals, Dallas, TX, USA) and total protein concentration was assayed using Alpha Diagnostics kit (Cat. No.: A6502-100; B6528-125). Other values were calculated using Pointe Scientific kits: triglycerides (T7531-150), glucose (G7519-100), cholesterol (C7509-400), HDL cholesterol (H7545-40), LDL cholesterol (L7574-40), alanine aminotransferase ALT (A7525-150), aspartate aminotransferase (ASP) (A7560-150), alkaline phosphatases (ALP) (A7516-120), and gamma-glutamyl transpeptidase (GGT) (G7571-120).

### 2.5. Statistical Analysis

The results of the animal experiment were analysed by one-way ANOVA. The significance of differences between the groups were calculated using the detailed Duncan’s test (to compare all the three groups) at *p* ≤ 0.05 and marked with letters, or applying the t-Student’s test (for comparison in pairs, raw vs. fermented) at *p* ≤ 0.05 (indicated by the *^,^** superscripts). Statistical analysis was performed using the SAS Enterprise Guide 5.1 computer program (Cary, NC, USA).

## 3. Results

Fermentation of lupine seeds by bacteria and yeast increased the CP and true protein contents (by 11.40% and 33.72%, respectively), similarly to CA (by 10.53%) and CF (by 13.75%) contents in comparison with RL (Table 2). On the other hand, in the fermented seeds, EE and NFE contents decreased in comparison with RL. RFOs were completely reduced in fermented seeds, whereas phytate phosphorus content decreased by about 34.4% after fermentation. Only total alkaloid contents and their composition were unchanged after fermentation. Contents of lysine, methionine and threonine in FL were slightly lower, whereas the level of cysteine was higher than in RL. Fermentation also strongly reduced the pH of the lupine product from about 5.50 to about 3.9.

The health status of the animals during the experiment was good. The average performance results are presented in Table 3. The replacement of 50% of SBM protein with raw or fermented lupine seeds in the diets for piglets did not significantly influence the examined indicators as compared to the SBM (*p* > 0.05). However, the ADBG and final body weight of animals offered FL were the highest, while there was no effect of fermentation on the performance results of pigs.

The effect of tested protein components in the diet on the physio-chemical indicators of the gastrointestinal tract status are presented in Table 4. No significant differences were found between the groups, but the highest dry matter content and ammonia content in the ileal digesta were found in the RL group (*p* > 0.05). Fermented lupine seeds in the diet significantly reduced acetic acid contents in the digesta as compared to the diet with raw seeds (*p* < 0.05). In this group, butyric acid was also found, whereas it was absent in the digesta of the other groups. Ammonia concentration in the caecum digesta was lower in the RL than in the SBM group (*p* < 0.05) (Table 4). In this part of the intestine, there were also no differences in the phenolic compound contents and the concentrations of most volatile fatty acids (*p* > 0.05). Fermented lupine meal in the diet caused a reduction in propionic acid contents compared to the RL group (*p* < 0.05), whereas in the SBM group a significantly higher (*p* < 0.05) content of isobutyrate was found compared to the other experimental groups.

In Colon I, the addition of fermented seeds to the diet significantly increased p-cresole content, but lowered total SCFA concentration as compared to the RL group (Table 5). In Colon I, isobutyrate was found only in the digesta of the SBM group. Generally, in Colon II, the pH of the digesta was statistically higher in the SBM group than in the other groups (*p* < 0.05), but the pH of the digesta of FL was higher than in the RL group (*p* < 0.05). Similarly, in the caecum and colon II, significantly lower concentrations of ammonia were found in the digesta of the RL group than in the SBM group (*p* < 0.05). There were also no differences between the groups in terms of phenolic compound contents in digesta of Colon II and III (*p* > 0.05). In Colon II, lower concentrations of acetic and propionic acids were recorded in the digesta of the SBM group compared to the RL group. Moreover, the isovalerate concentration in the digesta of the SBM group was higher than in the other groups. In Colon III digesta, higher ammonia concentrations were found in RL in comparison to FL. There were also no differences in bacterial enzyme activities and phenolic compound contents in the digesta of Colon III between the investigated groups (*p* > 0.05), with the exception of propionic acid.

The microbial composition of diets for pigs and the digesta are presented in Table 6. The microbiological status of the mixtures did not differ significantly (*p* > 0.05) except for the higher content of LAB in the FL diet (*p* < 0.05), where a significant effect of lupine fermentation was also found. In addition, there were no significant changes in the microbial contents of the small intestine and the caecum.

Substitution of SBM with RL and FL seed did not significantly affect the height of villi or the structural integrity of the mucosa (*p* > 0.05), In turn, in the group receiving lupine seeds in the diets, a significant reduction (*p* < 0.05) in the mucosa thickness was noted (Table 7). Fermentation significantly increased crypt depth (*p* < 0.05).

The mean results of the investigated biochemical parameters in the serum blood of pigs are presented in Table 8. The results did not differ significantly between the groups (*p* > 0.05). Only the contents of triglycerides in blood serum of pigs in the RL and FL groups were significantly higher than in the SBM group (*p* < 0.05). All the biochemical indicators of blood met the physiological standards.

## 4. Discussion

Fermentation is a natural process, in which microorganisms use and transform nutrients present in the biomass [25]. Generally, fermentation improves the nutritional value of feed components by increasing protein contents and the bioavailability of starch and structural carbohydrates, while also reducing some anti-nutritional compounds [16,17,26]. In the current research, fermentation increased crude protein contents by about 11% and true protein levels by about 33%, which was also observed by Kasprowicz-Potocka et al. [5] and Zaworska et al. [18]. The amino acid profile of protein was slightly changed after fermentation. The level of methionine was lower, but that of cysteine was higher in the fermented material. Similar observations were found in the study of Kasprowicz-Potocka et al. [5]. In turn, Feng et al. [27] found that an increase in the crude protein content in the case of fermentation mostly results from a decrease in the content of non-structural carbohydrates in the biomass, which was confirmed by this study. Similar observations were presented by Yabaya et al. [28], who fermented soybean cake with *S. cerevisiae.* During fermentation, yeast and bacteria also produce organic acids as a result of sugar degradation, which reduces the pH [25]. Similarly to this study, Yabaya et al. [28] also observed a decrease in pH from 5.6 to 5.1 during 24 h fermentation. Likewise, Kasprowicz-Potocka et al. [16] found a reduction in pH from 5.5 to 4.11 during fermentation of narrow-leafed lupine seeds and from 5.6 to 3.98 when fermenting yellow lupine seeds.

In plants rich in protein, including lupine, antinutritional substances are also present. Nowadays, sweet lupine seeds contain low alkaloid levels, so these substances are not a problem in animal nutrition. However, in legume seeds, phytic phosphorus is a major factor influencing the nutrition value of the feed. ANFs present in legumes are rather stable under heat treatment, but can be efficiently removed by fermentation [5,15,18]. In this study, fermentation by bacteria and yeast significantly decreased the level of phytic phosphorus and totally reduced RFOs, which was also found in other studies [18,26,29]. The reduction in these compounds may facilitate an increase in the level of lupine seeds in animal diets.

Many studies were provided to determine the nutritional value of different lupine seeds in fattening [1,3,5] and weaning pigs [4]. Some authors observed that sweet lupine is low in alkaloids; however, phytates and oligosaccharides are major factors influencing nutrient digestibility and thus reducing the growth performance of pigs [4,30]. There are very few studies using different processed lupine seeds as a feed component for pigs [18,31,32]. In the current work, the 50% substitution of SBM protein in diets with protein of raw or fermented narrow-leafed lupine seeds (20% and 18% seeds in the diet, respectively) did not affect body weight gains of pigs, feed intake and utilization in comparison to the control group offered a diet with soybean meal as the main protein component. In addition, Kasprowicz-Potocka et al. [4], by replacing 50% SBM protein with protein of raw and germinated narrow-leafed lupine seeds, found no significant differences between the groups, which is consistent with the results of this study. In the presented research, no positive effect of seed fermentation on rearing results of piglets was observed. Despite the more advantageous composition of the used product, the results in both experimental groups did not differ. Similar observations were recorded in the experiments on rats [17] in which, despite the use of seeds from improved lupine varieties, no improvement was reported in rearing results. The indicated differences in the discussed results may have been caused by the application of different lupine species in pig nutrition, as well as their different cultivars and different diet components and experimental designs.

Diets with lupine seeds (raw or fermented) affected, to a certain extent, the parameters of the small intestine and the caecum; however, they significantly affected the colon environment. There was a trend towards the lowest dry matter pH observed in the small intestine digesta of animals offered FL, which can suggest the best conditions for lactic acid bacteria proliferation. This may have also been the result of the favorable microbial status of fermented lupine, accompanied by a lower pH. In animals receiving RL, a higher production of acetate and butyrate in relation to the fermented seed group was found in the small intestine. This is consistent with previous observations [16,17,18]. Similarly to these authors’ research, Zdunczyk et al. [33] found that the amount of ammonia in caecal digesta produced during the bacterial degradation of lupine seeds in animals is lower than in the case of other feed components. Other studies have shown that fermented feed has an important potential to improve gut health and maintain gastrointestinal tract microbial homeostasis [12,34] and might possibly modulate the host gut microbiota. In contrast, some authors—similarly to the present study—failed to find beneficial effects of fermented feed on the microbial population and swine gut microbiota [8,35]. The main end products formed during the fermentation of non-digestible carbohydrates are mainly acetic, propionic, butyric and valeric acids, which was also confirmed by the current study. In the digesta from pigs fed with RL, higher concentrations of total SCFA were found in relation to the other groups (with the differences being significant only in Colon I). Higher SCFA concentrations in the ileal and colon contents of pigs fed a diet with RL compared with SBM and FL are likely to be the result of higher total amounts of non-starch polysaccharides and RFOs not being digested until the terminal ileum. The reduced concentration of SCFA may mean that the microbial activity in the digesta from animals fed with FL is partially inhibited; thereby, the introduced biomass constitutes a protein source in the initial sections of the gastrointestinal tract and does not show a probiotic effect in its subsequent sections [17]. Similarly, Zaworska et al. [18] showed that the use of raw lupine seeds affected SCFA concentrations, but fermented lupine seeds may reduce SCFA concentration in the ileal contents.

Gut microbiota plays a critical role in nutrient absorption, metabolism and host immune functions [36,37,38]. The composition of the intestinal content microbiota did not differ significantly with the small intestine and the caecum despite the observed deviations. A physiologically normal and specialized intestinal epithelium performs functions related to secretion, absorption and immune response [39]. The course, intensity of proliferation and cell growth are significantly influenced by environmental effects, first of all diet. In this study, significant differences were found in the histological structure of the small intestine epithelium. A thinner mucosa was observed in the villi of pigs in the group receiving raw lupine seeds in their diet, while deeper crypts were found in the group fed fermented seeds when compared to unprocessed seeds. This may indicate that the use of raw seeds in the diet may lead to changes, suggesting an abnormal development of the digestive tract. These results indicate disorders in the intensity of epithelial cell division in the group receiving RL. The development in villus height and crypt depth in the small intestine might be attributed to the role of the intestine epithelium as a natural barrier [40]. These results correspond to the findings of Rotkiewicz et al. [41], who in their experiments showed that a share of narrow-leafed lupine exceeding 25% in the feed mixture fed in the second period of fattening caused deformations or atrophy of intestinal villi in the duodenum and the jejunum of pigs. Similar results of the morphometric examination of the digestive tract were also given by Salgado et al. [42]. When evaluating the duodenum and the ileum, the cited authors showed a reduction in the height and width of villi, the depth and width of crypts, as well as the surface area in these sections. The values obtained by Salgado et al. [42] are consistent with the results presented in this study. This confirms the fact that the application of fermentation in the feed improvement process has a positive effect on the intestinal structure [8]. This effect of the fermentation process may be explained by the increased microbial activity in the digestive tract, which in turn affects the peristalsis and thus promotes an increase in villus height and crypt depth, while also contributing to the integrity of the mucosa. Moreover, butyric acid is the main source of energy for colonocytes, where it is used in proportions of up to 90% [43], thus it was probably used to build the mucosal epithelium in the SBM and FL groups.

The analysis of the lipid profile in blood serum did not reveal any significant disorders in pigs. The conducted analysis of biochemical blood parameters showed that the substitution of SBM protein in the mixed feed with lupine seeds (raw and fermented) did not change most of the investigated parameters except for the concentration of triglycerides. Similarly to this study, Prandini et al. [32] also reported no differences in the activity of liver enzymes in weaned piglets receiving raw and extruded seeds of white lupine and pea. In addition, Zraly et al. [30], when using white lupine seeds in fattener diets, found no differences in the biochemical parameters of blood. The results given by the above-mentioned authors fell within the physiological limits and corresponded to the results obtained in this study.

## 5. Conclusions

Fermentation of lupine seeds increased the shares of crude protein and true protein, while it also reduced the contents of anti-nutrients such as phytates and oligosaccharises, but did not reduce crude fibre content. The use of fermented seeds in the diet did not affect pigs’ performance, nor pig’s metabolic, microbial and most gastrointestinal tract parameters, but positively affected the ileal structure crypt depth. The concentration of p-cresole in the proximal segment of the colon and the ammonia content were reduced. The results also showed that the 50% substitution of SBM with lupine seeds (both raw and fermented) does not lead to a deterioration of the pigs’ performance results. Nevertheless, seed fermentation, due to the lack of a significant improvement in the values of these parameters, may prove to be economically unviable. Moreover, the considerable progress in breeding of new lupine varieties, characterized, e.g., by lower contents of anti-nutrients in seeds as well as seed improvement technologies, may contribute to a more extensive use of lupine seeds in nutrition of farm animals while at the same time causing no negative effects on the metabolism and digestive tract function in pigs. However, further research needs to be conducted in this respect under conditions found in large-scale farm production.

## Figures and Tables

**Table 1 animals-10-02084-t001:** The composition and nutritional value of diets.

Components %	SBM	RL	FL
Soybean meal (46%)	28.50	15.00	15.00
Narrow-leafed lupine meal	-	20.00	-
Fermented narrow-leafed lupine seeds	-	-	18.00
Wheat	49.80	41.05	43.60
Barley	15.35	16.95	16.95
Soya oil	2.50	3.00	2.50
Monocalcium phosphate	1.00	1.00	1.00
Limestone	1.40	1.40	1.40
L-Lysine 76%	0.40	0.50	0.50
DL-Methionine 99%	0.10	0.15	0.10
L-Threonine 98%	0.10	0.10	0.10
NaCl	0.35	0.35	0.35
Premix *	0.50	0.50	0.50
Calculated nutritional value (%)			
Dry matter	89.60	89.30	89.50
ME **(MJ/kg)	14.30	14.20	14.30
Crude protein	22.16	21.70	22.09
Ca	0.89	0.89	0.89
P	0.75	0.74	0.75
Lysine	1.26	1.29	1.29
Methionine	3.45	3.46	3.45
Threonine	7.38	7.40	7.40

* mineral and vitamin premix contained, per 1 kg; mg: choline chloride 40,000, Fe 15,000, Cu 4000, Co 60, Mn 6000, Zn 15,000, J 120, Se 30, antioxidants (butylated hydroxyanisole, butylated hydroxytoluene); IU: 1,500,000 vit. A, 300,000 vit. D3; mg: 10,500 vit. E, 220 vit. K3, 220 vit. B1, 600 vit. B2, 450 vit. B6, 1500 pantothenic acid, 3000 nicotinic acid, 300 folic acid; mcg: vit. B12 3700, biotin 15,000; g: Ca 260. ** ME—metabolizable energy; SBM—soybean meal; RL—raw lupine seeds; FL—fermented lupine seeds.

**Table 2 animals-10-02084-t002:** The chemical composition of raw and fermented lupine seeds.

Composition, g/kg in Dry Matter	RL	FL
Dry matter	899.4 ± 10.2	913.9 ± 8.6
Crude protein	354.6 ± 2.5	395.0 ± 1.2
Crude ash	39.9 ± 0.3	44.1 ± 0.2
Ether extract	56.3 ± 0.3	54.9 ± 0.2
Crude fibre	144.7 ± 0.7	164.6 ± 0.2
Nitrogen-free extractives	404.5 ± 1.8	341.3 ± 3.3
Total RFOs	65.3 ± 1.2	0.00 ± 0.00
Phytate phosphorus	5.30 ± 0.7	0.30 ± 0.2
Total alkaloids	0.20 ± 0.02	0.20 ± 0.10
Composition of alkaloids, in % of total alkaloids		
Angustifoline	7.42 ± 0.09	6.74 ± 0.11
Izolupanine	3.79 ± 0.07	3.86 ± 0.05
Lupanine	62.81 ± 0.98	60.90 ± 0.57
13OH-lupanine	25.99 ± 0.79	28.50 ± 0.88
Amino acids (g/100 g of protein)		
Lysine	5.31 ± 0.02	5.14 ± 0.04
Methionine	0.48 ± 0.01	0.45 ± 0.01
Cysteine	0.91 ± 0.04	1.12 ± 0.01
Threonine	4.11 ± 0.05	3.86 ± 0.03
pH	5.50 ± 0.05	3.90 ± 0.10

RL—raw lupine seeds; FL—fermented lupine seeds; RFOs—raffinose family oligosaccharides; results are expressed as means ± standard error.

**Table 3 animals-10-02084-t003:** Performance results of pigs.

Indicators	SBM	RL	FL	SEM	*p*-Value	
					Diet Effect	Fermentation Effect
Initial body weight (kg)	9.70	9.66	9.60	0.20	0.981	0.861
Final body weight (kg)	24.27	23.66	24.56	0.58	0.818	0.485
ADBG (kg)	0.540	0.519	0.554	0.020	0.683	0.352
FI (kg)/28 days	26.42	26.15	26.77	0.57	0.911	0.713
FCR (kg/kg)	1.85	1.86	1.82	0.05	0.922	0.692

SBM—soybean meal; RL—raw lupine seeds; FL—fermented lupine seeds; ADBG—average daily body weight gain; FI—feed intake; FCR—feed conversion ratio; SEM—standard error of mean.

**Table 4 animals-10-02084-t004:** Indicators of fermentation in fresh ileum, cecal digesta.

Parameters	SBM	RL	FL	SEM	*p*-Value	
					Diet Effect	Fermentation Effect
Ileum						
Dry matter (%)	10.75	13.45	10.84	1.12	0.121	0.074
pH	6.75	6.83	6.37	0.09	0.080	0.084
Viscosity (cP)	1.83	2.27	2.80	0.23	0.488	0.532
Ammonia (µM/g)	23.40	26.64	23.66	1.32	0.584	0.473
Acetate (μmol/g)	9.55 ^a,b^	15.97 ^a,^*	7.85 ^b,^**	1.49	0.045	0.021
Propionate (μmol/g)	BD	BD	BD	-	-	-
Butyrate (μmol/g)	BD	0.22	BD	-	-	-
Caecum						
Dry matter (%)	12.06	13.00	13.17	1.86	0.645	0.834
pH	5.56	5.33	5.41	0.09	0.602	0.434
Ammonia (µM/g)	22.95 ^a^	9.54 ^b^	14.59 ^a,b^	2.46	0.043	0.077
Phenol (μM/g)	0.09	0.07	0.26	0.05	0.253	0.226
p-Cresole (μM/g)	0.25	0.19	0.51	0.09	0.302	0.234
Indole (μM/g)	0.17	0.13	0.37	0.06	0.185	0.169
Acetate (μmol/g)	31.49	34.31	37.61	1.47	0.251	0.433
Propionate (μmol/g)	22.10	23.12 *	21.50 **	1.50	0.921	0.031
Isobutyrate (μmol/g)	0.22 ^a^	0.04 ^b^	BD	0.08	0.035	-
Butyrate (μmol/g)	8.88	10.00	7.84	0.78	0.571	0.284
Isovalerate (μmol/g)	0.11	BD	BD	-	-	-
Valerate (μmol/g)	1.96	1.22	0.20	0.35	0.376	0.285
Total SCFA (μmol/g)	64.78	68.68	67.67	3.65	0.236	0.320

SBM—soybean meal; RL—raw lupine seeds; FL—fermented lupine seeds; SCFA—short chain fatty acids; SEM—standard error of mean; BD—below detection; ^a^^,b^—means with different superscripts within a row are significantly different at *p* < 0.05 (ANOVA); *^,^**—data significantly different between raw and fermented lupine seeds at *p* < 0.05 (t-Student’s test).

**Table 5 animals-10-02084-t005:** Indicators of fermentation in fresh colon digesta.

Parameters	SBM	RL	FL	SEM	*p*-Value	
					Diet Effect	Fermentation Effect
Colon I						
pH	5.83	5.52	5.72	0.08	0.250	0.066
Ammonia (µM/g)	29.18	24.70	26.90	2.97	0.854	0.787
Phenol (μM/g)	0.01	0.010	0.01	0.00	0.650	0.348
p-Cresole (μM/g)	0.44	0.21 **	0.35 *	0.05	0.156	0.040
Indole (μM/g)	0.10	0.09	0.10	0.00	0.810	0.590
Acetate (μmol/g)	28.11	31.48	30.63	0.95	0.349	0.717
Propionate (μmol/g)	19.05	21.07	16.87	1.12	0.339	0.126
Isobutyrate (μmol/g)	0.27 ^a^	BD	BD	-	-	-
Butyrate (μmol/g)	9.59	11.54	9.24	0.68	0.362	0.271
Isovalerate (μmol/g)	BD	-BD	BD			
Valerate (μmol/g)	2.21	1.74	1.28	0.33	0.562	0.383
Total SCFA (μmol/g)	59.22	65.82 **	58.02 *	3.993	0.131	0.032
Colon II						
pH	6.07 ^a^	5.52 ^c,^**	5.72 ^b,^*	0.08	0.015	0.048
Ammonia (µM/g)	36.10 ^a^	20.58 ^b^	24.10 ^a,b^	3.04	0.046	0.594
Phenol (μM/g)	0.01	0.01	0.01	0.00	0.775	0.625
p-Cresole (μM/g)	0.60	0.39	0.42	0.06	0.255	0.787
Indole (μM/g)	0.09	0.09	0.08	0.01	0.492	0.459
Acetate (μmol/g)	12.91 ^b^	29.79 ^a^	27.45 ^a,b^	3.42	0.048	0.595
Propionate (μmol/g)	8.30 ^b^	20.16 ^a^	15.79 ^a,b^	2.23	0.035	0.165
Isobutyrate (μmol/g)	0.33	0.03	0.10	0.08	0.285	0.556
Butyrate (μmol/g)	5.40	12.14	9.75	0.19	0.470	0.319
Isovalerate (μmol/g)	0.50 ^a^	0.00 ^b^	0.00 ^b^	0.21	0.010	0.356
Valerate (μmol/g)	1.46	1.94	1.48	0.81	0.314	0.389
Total SCFA (μmol/g)	57.82	64.07	54.66	5.24	0.212	0.188
Colon III						
pH	6.14	5.86	5.90	0.07	0.175	0.773
Ammonia (µM/g)	34.26 ^a,b^	35.98 ^a,^*	23.11 ^b,^**	2.65	0.042	0.042
Phenol (μM/g)	0.01	0.01	0.01	0.00	0.134	0.270
p-Cresole (μM/g)	0.84	0.71	0.50	0.07	0.108	0.179
Indole (μM/g)	0.10	0.10	0.08	0.01	0.358	0.337
Acetate (μmol/g)	26.61	26.06	24.19	1.06	0.665	0.543
Propionate (μmol/g)	15.56	15.36	13.90	0.55	0.452	0.330
Isobutyrate (μmol/g)	0.83	0.49	0.42	0.09	0.090	0.627
Butyrate (μmol/g)	10.75	11.29	9.94	0.76	0.800	0.338
Isovalerate (μmol/g)	1.15	0.61	0.45	0.15	0.110	0.499
Valerate (μmol/g)	2.78 ^a^	2.21 ^a,b^	1.40 ^b^	0.25	0.050	0.131
Total SCFA (μmol/g)	57.68	56.02	50.30	3.26	0.325	0.398

SBM—soybean meal; RL—raw lupine seeds; FL—fermented lupine seeds; SEM—standard error of mean; I, II, III—colon segments; BD—below detection; ^a,b^—means with different superscripts within a row are significantly different at *p* < 0.05 (ANOVA); *^,^**—data significantly different between raw and fermented lupine seeds at *p* < 0.05 (t-Student’s test).

**Table 6 animals-10-02084-t006:** Microbial composition of diets with soybean meal, raw and fermented lupine and ileal and cecal digesta, log CFU/g.

log CFU/g	SBM	RL	FL	SEM	*p*-Value	
Diets					Diet Effect	Fermentation Effect
pH	5.61	5.50	5.27	0.10	0.221	0.320
Total bacteria number	5.72	5.61	5.91	0.35	0.153	0.213
Yeast and moulds	3.75	3.77	3.29	0.19	0.343	0.273
Lactic acid bacteria	2.70 ^b^	2.70 ^b,^**	6.81 ^a,^*	0.13	0.024	0.036
Coli group bacteria	5.01	5.54	4.60	0.12	0.944	0.787
Ileal digesta						
Total bacteria number	7.46	7.72	7.26	0.13	0.506	0.344
Caecal digesta						
Yeast and moulds	5.47	5.96	5.47	0.33	0.106	0.204
Lactic acid bacteria	7.84	8.62	9.46	0.47	0.150	0.302
Coli group bacteria	6.47	6.47	7.43	0.58	0.422	0.385

SBM—soybean meal; RL—raw lupine seeds; FL—fermented lupine seeds; SEM—standard error of mean; ^a,b^—means with different superscripts within a row are significantly different at *p* < 0.05 (ANOVA); *^,^**—data significantly different between lupine groups at *p* < 0.05 (t-Student’s test).

**Table 7 animals-10-02084-t007:** Morphometric parameters of ileum.

Parameters	SBM	RL	FL	SEM	*p*-Value	
					Diet Effect	Fermentation Effect
Mucosa thickness (μm)	697.52 ^a^	611.11 ^b^	676.93 ^a^	26.81	0.028	0.406
Height of villi (μm)	422.71	365.42	411.18	18.71	0.459	0.355
Crypt depth (μm)	330.08	287.78 **	308.06 *	16.89	0.638	0.032
SIM	1.34	1.33	1.50	0.08	0.847	0.655

SBM—soybean meal; RL—raw lupine seeds; FL—fermented lupine seeds; SIM—Structural integrity of the mucosa; SEM—standard error of mean; ^a^^,b^—means with different superscripts within a row are significantly different at *p* < 0.05 (ANOVA); *^,^**—data significantly different between raw and fermented lupine seeds at *p* < 0.05 (t-Student’s test).

**Table 8 animals-10-02084-t008:** Biochemical blood indexes in piglets.

Parameters	SBM	RL	FL	SEM	*p*-Value	
					Diet Effect	Fermentation Effect
TG (mg/dL)	50.14 ^b^	58.19 ^a^	57.62 ^a^	3.02	0.045	0.949
Glucose (mg/dL)	101.61	93.39	97.59	3.47	0.671	0.706
Cholesterol (mg/dL)	77.60	76.50	79.23	3.37	0.955	0.755
HDL cholesterol (mg/dL)	31.81	42.48	59.55	5.01	0.056	0.176
LDL cholesterol (mg/dL)	30.20	28.84	26.39	2.43	0.839	0.739
NEFA (mmol)	0.40	0.42	0.41	0.00	0.195	0.240
TP (g/dL)	6.10	6.13	6.38	0.15	0.767	0.528
ALT (IU/l)	38.68	21.33	20.88	3.78	0.875	0.918
AST (IU/l)	19.40	18.92	16.18	1.87	0.785	0.651
ALP (IU/I)	92.95	91.31	91.58	4.60	0.990	0.78
GTP (IU/I)	17.41	19.07	28.19	2.83	0.267	0.193

SBM—soybean meal; RL—raw lupine seeds; FL—fermented lupine seeds; TG—Triglycerides; HDL—high-density lipoprotein cholesterol; LDL—low-density lipoprotein cholesterol; NEFA—nonesterified fatty acids; TP—total protein; ALT—Alanine aminotransferase; AST—Aspartate aminotransferase; ALP—alkaline phosphatases; GTP—gamma-glutamyl transpeptidase; SEM—standard error of mean; ^a,b^—means with different superscripts within a row are significantly different at *p* < 0.05 (ANOVA).

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
