# Peer review of "Growth Performance, Gut Environment and Physiology of the Gastrointestinal Tract in Weaned Piglets Fed a Diet Supplemented with Raw and Fermented Narrow-Leafed Lupine Seeds"

_animals, 2020, doi:10.3390/ani10112084_

Round 1
Reviewer 1 Report
- The purpose of the study is not properly formulated, such formulation is more appropriate for the tasks.
- I do not think that 8 animals in a group is a sufficient number of animals to determine the impact on growth performance.
- In “Material and methods” - needs to describe more precisely which animals were used in the study. Only in the Abstract can be found that in the study were used “24 male pigs”. - The same term should be used to describe the growth rate, now it called differently: line 104 - average daily body gain (ADBG), line 191 and table 3 - daily weight gain.
- Is not clear what content of SBM was in the control diet: line 94 states that 28.5.% and in Table 1 it was 28%.
- In Table 1, it would be useful to include the content of dry matter, methionine, threonine (in calculated nutrition value).
- line 95 - incorrect “final diet”, probably should be an “experimental diet”.
- Table 2 provides the chemical composition of lupins in %. When analyzing data, percentages are calculated from percentages. This is not a suitable calculation, it would only be appropriate when the data are presented in absolute terms (g, kg, mg, etc.). In addition, some of these values are even incorrect, for example: line 181 – is written 44.4%, although the calculation gives 43.4%.
- In Table 2, it may be useful to indicate the content of dry matter and energy.
- Line 201 - should be Table 4 (not 5).
- The analysis of data in Table 5 is not clear and inconsistent. Jumping between Colon II, III, I. It is unclear whether line 220 really refers to caecum (Table 4) or Colon I, II.
- Line 248 - it should be specified that biochemical parameters in the serum of blood.
- Weak introduction, the literature review should be strengthened.
Author Response
We would like to thank the Reviewers for their comments and for considering a revised version of our manuscript. All changes made in the manuscript are marked in red. The manuscript in the revised form has been approved by all the co-authors.
Reviewer 1
- The purpose of the study is not properly formulated, such formulation is more appropriate for the tasks.
Our response: Thanks for this comment. The aim of the study was revised. L.: 81-85
- I do not think that 8 animals in a group is a sufficient number of animals to determine the impact on growth performance.
Our response: These number is enough for statistic design. Our team previously published many researches on growing pigs where the number of repetitions ranged from 6 to 10 and none of the reviewers questioned this number. Also, other reviewers of this manuscript had no objections.
For examples articles:
- Zaworska-Zakrzewska, M.Kasprowicz-Potocka, M. Twarużek, R. Kosicki, J. Grajewski, Z. WiÅ›niewska, A. Rutkowski. A Comparison of the Composition and Contamination of Soybean Cultivated in Europe and Limitation of Raw Soy Seed Content in Weaned Pigs’ Diets . Animals 2020, 10, 1972; doi:10.3390/ani10111972.
- Nowak, P., Kasprowicz-Potocka, M., Zaworska, A., Nowak, W., Stefańska, B., Sip, A., Grajek W., Juzwa W., Taciak M., Barszcz M., Tuśnio A., Grajek K., Foksowicz-Flaczyk J., Frankiewicz A. (2017). The effect of eubiotic feed additives on the performance of growing pigs and the activity of intestinal microflora. Archives of Animal Nutrition, 71(6), 455-469. https://doi.org/10.1080/1745039X.2017.1390181.
- Kasprowicz-Potocka M., Zaworska, A., Kaczmarek S. A., Hejdysz M., Mikuła R., Rutkowski A. (2017). The effect of Lupinus albus seeds on digestibility, performance and gastrointestinal tract indices in pigs. J Anim. Phys Anlm. Nutr. 101.5 (2017): e216-e224. DOI: 10.1111/jpn.12594.
- Kasprowicz-Potocka M., Zaworska, A., Kaczmarek S. A., Rutkowski A. (2016). The nutritional value of narrow-leafed lupine (Lupinus angustifolius) for fattening pigs. Archives of Animal Nutrition, 70(3), 209-223.
- In “Material and methods” - needs to describe more precisely which animals were used in the study. Only in the Abstract can be found that in the study were used “24 male pigs”.
Our response: We followed this suggestion and some details about animals were added in section M&M. L.: 97 “The experiment was conducted on 24 castrated male weaned piglets (P76 × Naima), aged 35 days…”
- The same term should be used to describe the growth rate, now it called differently: line 104 - average daily body gain (ADBG), line 191 and table 3 - daily weight gain.
Our response: We followed this suggestion and changed in the whole manuscript.
- Is not clear what content of SBM was in the control diet: line 94 states that 28.5.% and in Table 1 it was 28%.
Our response: We agree. The control diet contained 28.5% of SBM. We changed data in table 1.
- In Table 1, it would be useful to include the content of dry matter, methionine, threonine (in calculated nutrition value).
Our response: in table 1 we added the content of dry matter, methionine, threonine.
- line 95 - incorrect “final diet”, probably should be an “experimental diet”.
Our response: We followed this suggestion and corrected in text L. 104
- Table 2 provides the chemical composition of lupins in %. When analyzing data, percentages are calculated from percentages. This is not a suitable calculation, it would only be appropriate when the data are presented in absolute terms (g, kg, mg, etc.). In addition, some of these values are even incorrect, for example: li nne 181 – is written 44.4%, although the calculation gives 43.4%.
Our response: We agree with the Reviewer, and we changed data in table 3., In text (L. 181) it was mistake- we corrected calculation.
- In Table 2, it may be useful to indicate the content of dry matter and energy.
Our response: We added the content of dry matter in raw and fermented lupine seeds. We did not analyze energy in both components.
- Line 201 - should be Table 4 (not 5).
Our response: It was done. L. 211
- The analysis of data in Table 5 is not clear and inconsistent. Jumping between Colon II, III, I. It is unclear whether line 220 really refers to caecum (Table 4) or Colon I, II.
Our response: The joint analysis of data for the three parts of colon was to make a synthetic description not to extend the chapter Results. Yes, line 220 really refers to caecum and Colon II (added in text), in Colon I no differences were found.
- Line 248 - it should be specified that biochemical parameters in the serum of blood.
Our response: Some details were added (L. 259)
- Weak introduction, the literature review should be strengthened.
Our response: Introduction was changed and some part added with new literature according to the Reviewer instruction (L. 52-78).
Reviewer 2 Report
GENERAL COMMENTS:
This paper is interesting and practical, therefore, deserve for publication.
SPECIFIC COMMENTS:
Table 2. – Give information about dry matter content.
In the table 4 and 5 sometimes “0.00” is presented with letters indicating significant differences – this is confusing in my opinion: is that mean that value 0 was considered for data evaluation? If so, this is not justified herein. Though, if there is a value behind this different than 0, I would recommend to report this data as “trace level” and give the true value of it in the table footnote.
The work needs improvement according to the comments. After taking these into account, I recommend it for publication in the Animals.
Sincerely

Author Response
We would like to thank the Reviewers for their comments and for considering a revised version of our manuscript.
Reviewer 2
- This paper is interesting and practical, therefore, deserve for publication.
Our response: Thank you for a positive opinion.
- Table 2. – Give information about dry matter content.
Our response: We followed this suggestion and added dry matter content (table 2).
- In the table 4 and 5 sometimes “0.00” is presented with letters indicating significant differences – this is confusing in my opinion: is that mean that value 0 was considered for data evaluation? If so, this is not justified herein. Though, if there is a value behind this different than 0, I would recommend to report this data as “trace level” and give the true value of it in the table footnote.
Our response: We followed this suggestion and changed in table 4 and 5 and in text L. 20 and 235-236.
- The work needs improvement according to the comments. After taking these into account, I recommend it for publication in the Animals.
All changes made in the manuscript are marked in red. The manuscript in the revised form has been approved by all the co-authors.
Round 2
Reviewer 1 Report
Dear Authors,
It is good that you have accepted many of the comments and corrected them. Now the quality of the article has really improved.
In response to your comment on the number of animals, I can note that of course, I understand well that the study was carried out with this number of animals you have indicated, which unfortunately cannot be corrected now.
Even a smaller number of animals is sufficient for statistical design.
However, if you have a goal to determine the influence of fermentation on growth performance, this simple indicator of animal productivity should be assessed more solidly.
Usually, less animals are used to determine physiological parameters (which are often invasive and expensive), and productivity is assessed on the basis of more animals.
Also, if you had included more animals in the study, probably you could get more accurate data and possibly even sinificant differences.